# Do CEO Duality and Ownership Concentration Impact Dividend Policy in Emerging Markets? The Moderating Effect of Crises Period

Anis El Ammari 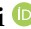

Department of Accounting and Finance, Faculty of Economic Sciences and Management, University of Monastir, Mahdia 5111, Tunisia; ammari_anis1@yahoo.fr; Tel.: +216-982-576-72

**Abstract:** Despite developments of recent theoretical and numerous empirical studies on the policies effectively adopted by companies, the dividend distribution policy (DDP) remains largely unexplained. In this regard, the main purpose of the current study is to empirically examine the effects of both CEO duality and ownership concentration on DDP during a crisis period. Furthermore, we test, using an interaction variable, the moderating effect of the crisis period on the association between both the degree of CEO duality and the ownership concentration on the DDP by analyzing panel data on selected listed firms in an emerging economy, namely, Tunisia. Based on a sample made up of 576 firm-year observations over the period 1996–2019, the findings of this research indicate that the crisis period plays an important role in mitigating the positive effect of both CEO duality and ownership concentration on DDP. The findings confirm furthermore that the crisis period on the one hand and both CEO duality and ownership concentration on the other represent two competing forces influencing DDP. Our results also support the agency theory on which DDP depends, among other things, family ownership, board and company size, and ROE.

**Keywords:** CEO duality; ownership concentration; dividend distribution policy; crisis period; moderating effect

## 1. Introduction

The dividend is a fundamental component of profitability, which makes it possible on the one hand to retain shareholders who are interested in a high income and, on the other hand, to convey information between managers, shareholders, the company, and the market (Floyd et al. 2015; Brawn and Sevic 2018). In this respect, the issue of DDP has been the subject of numerous theoretical and empirical studies for several decades (e.g., Lintner 1956; Miller and Modigliani 1961; Rozeff 1982; Easterbrook 1984; Smith and Watts 1992; Adaoglu 2000). Researchers have mainly focused on the different economic and financial elements that can guide companies' choices in terms of dividend distribution. Nevertheless, the debate on the DDP has caused much ink to flow to this day without reaching unanimous conclusions regarding the factors affecting this dividend policy (e.g., Tahir et al. 2014; Benjamin and Zain 2015; Yarram and Dollery 2015; Al-Najjar and Kilincarslan 2016; Shahid et al. 2016; Sindhu et al. 2016; Chen et al. 2017; Mehdi et al. 2017; Obaidat 2018; Sumail 2018; Sanan 2019; Ye et al. 2019; Jabeen and Ahmad 2019; Ahmad et al. 2019). According to Ahmed et al. (2020), dividend policy is a "puzzle" whose determinants are still poorly understood. At the same time, the increase in additional research work on the DDP—like Taleb and Ben Lahouel (2020); Duqi et al. (2020); Tahir et al. (2020a, 2020b); Endang et al. (2020); Grey et al. (2020); Juhmani (2020); Ahmed et al. (2020); Thompson and Adasi Manu (2021); Ain et al. (2021); Bataineh (2021); Dissanayake and Dissabandara (2021); Hasan et al. (2021); Trinh et al. (2021); El Ammari (2021) and even

other authors—justifies the fact that this research topic is a topical subject and that the debate on the reality of this famous financial phenomenon is not yet closed and still requires heated debate.

Although not yet solved, a wave of competitive theories emerged in order to understand and explain firms' DDP behavior (e.g., Jensen 1986; Shleifer and Vishny 1997; Al-Malkawi et al. 2010). From a theoretical point of view, agency theory provides a satisfactory analytical framework for explaining this issue. To this end, the present study relies on the agency theory to explain the DDP problems. Indeed, Shleifer and Vishny (1997) argue that majority shareholders prefer to derive private benefits from the firm and tend to favor a limited distribution of dividends. According to both authors, this is a form of expropriation of minority shareholders. In this respect, to the extent that minority shareholders are unable to force majority shareholders to distribute dividends, Du Boys (2008) argues that for shareholders with substantial control over the firm, further reducing dividend distribution allows them to obtain private profits and thus increase their personal wealth.

On the other hand, based on the agency theory perspective, only a handful of recent studies (e.g., Chen et al. 2017; Sumail 2018; Sanan 2019; Tahir et al. 2020c; Graham et al. 2020; Thompson and Adasi Manu 2021; Dissanayake and Dissabandara 2021) looked at the board of directors' characteristics as a key factor that could affect DDP. From this angle, special attention is drawn to the topic of CEO duality in the dividend policy literature; but still with no tangible evidence confirming the impact of this duality on dividend payouts (Baliga et al. 1996). Indeed, CEO duality shows the corporate board power structure and represents one of its main efficiency variables.

While many efforts were made to justify DDP, very few attempts were directly related to political, financial, and/or economic crises (Mehdi et al. 2017). In this logic, we find it relevant to seize the opportunity offered through the last political crisis due to the Tunisian revolution that occurred in 2011 to reconsider these researches in a context of generalized instability and strong turbulence. According to Mehdi et al. (2017) and Taleb and Ben Lahouel (2020), these political, financial, and/or economic crises have clear consequences on the DDP. In this sense, Suwaidan and Khalaf (2020) claim that dividend payment was deemed necessary to attract capital during a transition period.

Thus, insofar as CEO duality and ownership concentration vary significantly over time, this study aims to empirically study the effects of both CEO duality and ownership concentration on DDP. Furthermore, like Zulfikar et al. (2020) and Ngatno et al. (2021), we test, using an interaction variable, the moderating effect of the crisis period on the association between both the degree of ownership concentration and the DDP on the one hand, and between the CEO duality and the DDP on the other.

For this purpose, notwithstanding the presence of relatively abundant literature on DDP, there are, however, relatively few empirical studies that accurately and clearly detect the combined effect of ownership concentration and CEO duality on DDP, especially in emerging countries (Suwaidan and Khalaf 2020; Thompson and Adasi Manu 2021) and more particularly during a crisis period (Mehdi et al. 2017). In this sense, our study particularly focuses on an emerging market, namely the Tunisian context, which still remains insufficiently investigated. Our paper's interest lies in particular in its consideration of the political and financial crisis period caused by the 2011 Tunisian revolution. In this country, the companies are precisely characterized by extensively concentrated ownership and weak corporate governance compared with other developed countries (Gana and El Ammari 2013). The understanding of this observation can therefore help solve the famous "dividend puzzle" and better account for the functioning and specificities of Tunisian companies since Tunisia continues to attract and increase foreign investments in this crisis period after the revolution and especially during the worldwide COVID-19 pandemic.

The major contributions of this paper to the existing literature are as follows. Firstly, to the best of our knowledge, this is the first study that examines exclusively the combined factors, such as CEO duality and ownership concentration, on DDP by analyzing panel data

on selected listed firms in an emerging economy. Secondly, another interesting contribution is that our paper tests this association mainly during crises periods. Thirdly, the originality and the relevance of our study consist in studying the moderating effect of the crisis period on the association between the degree of ownership concentration and DDP on the one hand, and between the CEO duality and DDP on the other. To the author's knowledge, this is a pioneer study that tests the crisis period as a moderating variable. Lastly, this study also presents several theoretical and empirical contributions.

Taken together, our results highlight significant relationships between ownership concentration, CEO duality, and DDP. More specifically, the findings provide substantive evidence that the political crisis plays an important role in mitigating the positive effect of ownership concentration and CEO duality on DDP. Our results also support the agency theory on which dividend distribution depends, among other things, family ownership, board and company size, and ROE.

To this end, the remainder of this paper is organized as follows. The literature review and the formulation of research hypotheses are developed in Section 2. Section 3 describes the research design. Section 4 presents the empirical results and discussion. Finally, Section 5 provides the conclusion and the implications of the paper.

## 2. Literature Review and Hypothesis Development

### 2.1. Ownership Concentration and Dividend Distribution Policy

The ownership structure is a determinant factor of company policies. The importance of majority shareholders affects the decision-making power in their favor, such as dividends decisions (Gugler 2003; Kouki and Guizani 2009). According to the agency theory, there are globally two divergent views regarding the effect of ownership concentration on DDP. On the one hand, Shleifer and Vishny (1986) argue that concentrated ownership encourages majority shareholders to exercise greater supervision over the management of the firm. In this regard, Easterbrook (1984) indicates that majority shareholders demand high dividends for two reasons: to resolve the agency conflict and to recover the supervision costs. According to Jensen (1986), the payment of dividends reduces agency costs by limiting the liquidity available for managers' discretionary spending. Similar results are obtained by Renneboog and Szilagyi (2020). Their sample covers Dutch firms listed on Euronext Amsterdam and the new market NMAX. They find that ownership concentration, with the objective of limiting FCF agency costs, enhances dividend payout.

At the same time, Kouki and Guizani (2009) find that Tunisian firms with concentrated capital ownership distribute more dividends. Such outcomes meet those of Faccio et al. (2001). The latter stipulated that a second large shareholder presence boosts dividend payment. This positive relationship is confirmed in the Moroccan context by the results of Mossadak et al. (2016) who show that ownership concentration implies higher dividend payments. In the same context, another more recent study supports the findings of Mossadak et al. (2016), including Boujjat et al. (2017). The authors demonstrated the positive role of capital concentration in determining the level of dividend distribution. Likewise, Thanatawee (2013) tackles the dividend policy and ownership structure relationship in the Thai context. The researcher empirically proves that a high ownership concentration and a large number of shareholders in a firm increase its dividend payment. The researcher also stipulates that a firm with more equity pays higher dividends to its shareholders.

In the same way, the findings of Shehu (2015) in the Malaysian context show that concentrated ownership is found to be positive and significant in influencing the dividend payout. The results obtained by the author are consistent with the conclusions by Easterbrook (1984) and Shleifer and Vishny (1997) who argue that in a situation of ownership the principal shareholders require a high level of dividends in order to reduce agency costs. More recently, Anh and Tuan (2019) and Arora and Srivastava (2021) also prove that ownership concentration plays a vital role in the dividend payment policy in the Vietnamese and Indian contexts, respectively. They prove that deciding to pay dividends or not is influenced by the level of ownership concentration structure. By examining

the relationship between ownership concentration and dividend payout in India, Arora and Srivastava (2021) found that ownership concentration is positively correlated with dividend payout. The authors add that the presence of a large shareholder, other than an individual, outside the promoter group has a negative influence on the dividend payout. This negative influence is higher for a financial company and is strongly dependent on the size of the shareholding of such a large shareholder relative to that of the promoter group.

Furthermore, Shafai and Shafai (2020) look at firms in Malaysia. Based on their empirical results, the authors confirm a positive and significant relationship related to concentrated and foreign ownership structure in Malaysia. Indeed, firms assume that foreign investors are more inclined to the firm's active monitoring in order to mitigate the agency issues. This enables the firms to have higher dividend payouts. Additionally, concentrated ownership is likely to boost a firm's dividend payout since largely concentrated shareholders control the firm, this reduces agency problems and is in line with the agency theory.

On the other hand, other authors (e.g., Jensen 1993; Chen et al. 2005; Ghosh and Sirmans 2006) find that ownership concentration is negatively associated with DDP. According to them, these results are consistent with agency models in which dividends represent a substitute for poor monitoring by a firm's shareholders (Boubaker and Nguyen 2014). For this purpose, Maury and Pajuste (2003) argue that dividend payout is reduced since ownership concentration reduces a firm's value. Moreover, in the German context, Gugler and Yurtoglu (2003) advance that majority shareholders could gain private benefits to the detriment of minority shareholders due to a low dividend payout. Rahman (2002) stipulates that dividend smoothing is negatively related to a country's ownership level as well as firm concentration. In another similar study, Shahid et al. (2016) confirm these findings in the Pakistani and Indian contexts. They find a negative link between ownership concentration and DDP. In another context, Sumail (2018) examines a sample of companies listed in the Indonesia Stock Exchange. Their results show that ownership concentration is negatively associated with the dividend payout rate.

For their part, Anh and Tuan (2019) examine the potential association between ownership structure and listed firms' DDP in Vietnam. According to their results, the level of ownership concentration structure positively affects Vietnamese firms' decision of paying dividends, namely for firms listed on the HNX stock market. At the same time, Gyapong et al. (2019) consolidate these results in the Australian context. They demonstrate that when ownership concentration is high, board gender diversity reduces dividend payments.

More recently, Taleb and Ben Lahouel (2020) detect a negative and significant relationship between ownership concentration and a dividend payout ratio in the Tunisian context. According to the authors, Tunisian firms characterized by concentrated ownership prefer to retain profits rather than pay them out as dividends. Based on the previous discussion, we present our first testable hypothesis as follows:

**Hypothesis 1 (H1).** *There is a significant association between ownership concentration and Tunisian firms' dividend distribution policy, ceteris paribus.*

### 2.2. CEO Duality and Dividend Distribution Policy

Board duality, i.e., when the roles of Chairman of the Board and Chief Executive Officer are held by the same person, is another characteristic of board effectiveness. In this regard, many studies shed light on CEO duality, but there is no conclusive proof of its impact on dividend payouts (Baliga et al. 1996). Indeed, CEO duality is an indicator of the board's power structure and an important variable of the board's efficiency. For this purpose, Jensen (1993) recommends separating the position of Chairman of the Board from that of management to reduce the CEO's discretionary power and ensure the effectiveness of the Board of Directors. When the chief executive officer is the same person who holds the position of chairman of the board of directors or the supervisory board, the former acquires sufficient influence over the operations of the latter and makes it unable to perform its key

functions effectively. Several governments reports stress the importance of the board of directors functioning at arm's length from senior management. Indeed, the CEO and the Chairman of the Board have different roles. The combination of these two roles constitutes a strong concentration of power that may call into question the independence of the board, which will have negative consequences on the shareholders' wealth (Baliga et al. 1996). This results in a weak control system that could negatively affect the distribution of dividends to shareholders. Such an outcome meets the agency theory arguments in the sense that this duality strengthens CEO entrenchment and hinders the board monitoring efficiency.

On the other hand, Arshad et al. (2013) found a negative and significant relationship between the duality of leadership style and dividend distribution. At the same time, Asamoah (2011) reveals that the CEO duality acts negatively both on the decision to pay dividends and on the amount of dividend paid. As for Chen et al. (2005), family businesses often recur to CEO duality, which limits controlling shareholders' power and has a negative effect on the business's financial performance and dividends allocation. Likewise, Shehu (2015) find also that the independent non-executive director is found to be significant in influencing the dividend payout in a negative direction. Recently, using Brazilian and Chilean family firms, Turrent et al. (2020) show that the combination of the functions of chairman and CEO has a negative effect on the distribution of dividends. At the same time, Suwaidan and Khalaf (2020) examine the impact of ownership structure on the DDP in a sample of manufacturing firms listed on the Amman Stock Exchange. Based on the multiple regression analysis outcomes, CEO duality is negatively associated and statically significant with the dividend per share variation. More recently, Thompson and Adasi Manu (2021) studied a sample comprising U.S. firms. The authors found that the board chairman's independence and the directors' voting rights have a negative and significant impact on the probability of dividend distribution.

Based on this analysis, it is expected that boards chaired by the CEO will be less effective in limiting managerial discretion, particularly with respect to the DDP. To this end, in order to overcome the poor governance practices adopted by the management team members, other authors (e.g., Shahid et al. 2016; Sumail 2018; Tahir et al. 2020a) stipulate that the duality of functions would favor the distribution of dividends for controlling shareholders in firms with concentrated ownership. In this sense, Taleb and Ben Lahouel (2020) recently found that Tunisian firms with a CEO in charge tend to pursue a generous DDP. Similarly, this positive and statistically significant association was also corroborated by Mehdi et al. (2017) based on a sample of selected companies from East Asian countries and the Gulf Cooperation Council. According to the authors, firms, where the two positions of CEO and chairman are held by the same person, are more likely to adopt a higher dividend payout policy. The researchers' argument here is that this duality in emerging markets is not an efficient means to account for the risk of expropriation. That is why shareholders demand higher dividend payouts to solve free cash flow issues. Such an explanation confirms the results of Bradford et al. (2013) who believe that, in developing markets, CEO duality is not an efficient control instrument. More recently, Tahir et al. (2020a) conclude, using a sample of Malaysian firms, that CEO duality has a positive and a statistically significant impact on DDP.

However, some research studies have found no significant relationship between CEO duality and DDP. In this sense, using non-financial companies listed on the Malaysian Stock Exchange, Tahir et al. (2020b) confirm the existence of a strong negative but statistically insignificant effect between CEO duality and DDP. In parallel, by investigating the role of corporate governance in the dividend decision of non-financial companies listed on the Colombo Stock Exchange of Sri Lanka, Nazar (2021) confirms that the CEO duality showed an insignificant negative impact on the dividend payout ratio.

All of these arguments allow us to advance our second hypothesis:

**Hypothesis 2 (H2).** *There is a significant relationship between CEO duality and Tunisian firms' dividend distribution policy, ceteris paribus.*

*2.3. Dividend Distribution Policy in Crisis Period: Moderating Effect*

An important line of research that has been developing in recent years, particularly in the wake of the 2008 financial crisis, is to test the relationship between the financial policies adopted by firms and governance mechanisms in a crisis period (Boubaker et al. 2012). In this sense, Hauser (2013) shows that the proportion of firms distributing dividends declined between 2008 and 2009, and the DDP varied considerably during the crisis. Furthermore, by analyzing a sample of large companies listed on the exchanges of 10 Central and Eastern European countries, Lace et al. (2013) show that payout ratios declined slightly during the 2008–2009 financial crisis. Similar results are obtained by Reddemann et al. (2010). Their findings indicate that, during the recent 2008–2009 financial crisis, German and European insurance companies did adjust their distribution policy through dividend suspension payments in order to strengthen their liquidity and preserve their capital base.

For their part, using the 2008 financial crisis as an analytical framework, Sawicki (2009) and Mehdi et al. (2017) prove that there is a positive relationship between the governance mechanisms adopted by firms and the level of profit distribution during a crisis. Specifically, these studies show that better-governed firms pay higher dividends in crisis periods and vice versa. On their side, using a sample of Australian companies, Gyapong et al. (2019) also indicate that the financial crisis period was associated with high dividend payments; however, women directors restrained the payment of dividends during the crisis period. This relationship was most recently tested by Taleb and Ben Lahouel (2020) in the Tunisian context. These authors detect a negative and significant relationship between the level of dividend distribution and the crisis period. The authors explain this negative relationship by the fact that during the crisis, board members become more risk-averse. Based on this information, we formulate the following hypothesis:

**Hypothesis 3 (H3).** *The crisis period has a negative connection with DDP while moderating the effects of both ownership concentration and CEO duality.*

### 3. Research Design
*3.1. Conceptual Framework*

The study aims to examine the effects of both ownership concentration and CEO duality on DDP in a crisis period. Furthermore, this research tests, using an interaction variable, the moderating effect of the crisis period on the association between both the degree of ownership concentration and the CEO duality on the DDP. The DDP of this study was measured by two dependent variables: Dividend yield and Dividend payout ratio. Two models were applied by the researchers. These models will be illustrated later in this section. Figure 1 below illustrates the conceptual framework for the associations explored in this study.

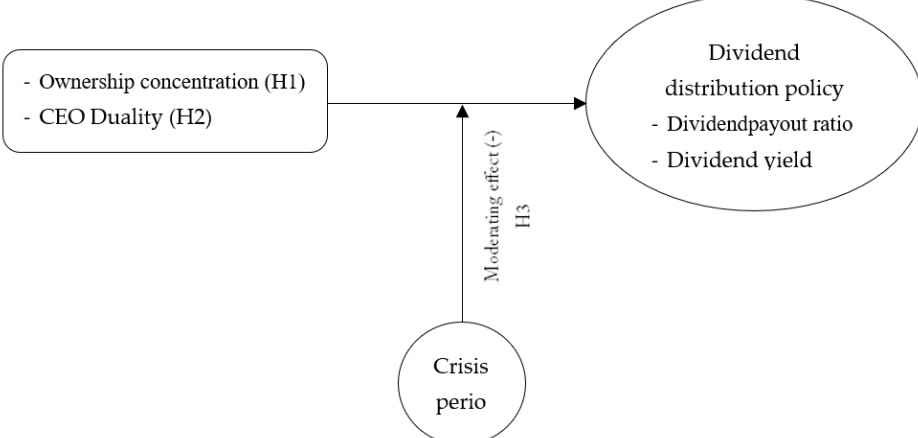

**Figure 1.** Conceptual framework.

*3.2. Models Specification, Definitions, and Measurements of Variables*

　　　To test the empirical validity of the hypotheses formulated above, we then estimate two-panel data equations which report the values of the variables considered for a set of 576 observations from 24 companies listed on the Tunis Stock Exchange (BVMT) observed over the 1996–2019 period. Dormont (2002) indicates that this double dimension (individual and temporal) makes it possible to expand the number of observations and the degree of freedom and simultaneously takes into account the dynamics of behavior and their possible heterogeneity. To this end, the following model is performed:

$$DDP_{it} = \alpha_0 + \alpha_1 Own\_C_{it} + \alpha_2 Board\_D_{it} + \alpha_3 Own\_M_{it} + \alpha_4 Own\_F_{it} + \alpha_5 Board\_S_{it} + \alpha_6 Board\_W_{it} + \alpha_7 Firm\_S_{it} + \alpha_8 ROE_{it} + \alpha_9 Lev\_R_{it} + \varepsilon_{it} \qquad (1)$$

whereby:
-  Dependent variables:

　　　In the framework of this study, we use two different dependent variables to define the dividend distribution policy (DDP) namely:

　　　Div_PR = the dividend payout ratio;

　　　Div_Y = the dividend yield;
-  Moderating variable:

　　　Cris_P = the crisis period;
-  Test variables:

　　　Own_C = the ownership concentration;

　　　Board_D = the CEO Duality;
-  Control variables:

　　　Own_M = the managerial ownership;

　　　Own_F = the family ownership;

　　　Board_S = the board size;

　　　Board_W = the women's presence on boards;

　　　Firm_S = the firm size;

　　　ROE = the return on equity;

　　　and Lev_R = the leverage ratio.

　　　In order to capture the incremental effect of the political crisis and economic slowdown due to the 2011 Tunisian revolution in accounting for firms' DDP, we run our basic model by adding a new variable (Cris_P) to the regressions. The variable Cris_P takes the value of 1 for the 2010–2012 period and 0 otherwise. Thus, model 2 is performed after introducing this variable as follows:

$$DDP_{it} = \alpha_0 + \alpha_1 Cris\_P_t + \alpha_2 Own\_C_{it} + \alpha_3 Board\_D_{it} + \alpha_4 Own\_M_{it} + \alpha_5 Own\_F_{it} + \alpha_6 Board\_S_{it} + \alpha_7 Board\_W_{it} + \alpha_8 Firm\_S_{it} + \alpha_9 ROE_{it} + \alpha_{10} Lev\_R_{it} + \varepsilon_{it} \qquad (2)$$

　　　To test for the moderating effect of the crisis period (Cris_P) on both the relationship between ownership concentration and DDP and the relationship between CEO duality and DDP, the overall sample is sub-grouped into three sub-samples, namely: period before crisis, period during crisis, and period after crisis. To test H3, we regress model 1 for the three groups considered. Accordingly, model 3 is performed as follows for the crisis period:

$$DDP_{it} = \alpha_0 + \alpha_1 Own\_C_{it} + \alpha_2 Board\_D_{it} + \alpha_3 Own\_M_{it} + \alpha_4 Own\_F_{it} + \alpha_5 Board\_S_{it} + \alpha_6 Board\_W_{it} + \alpha_7 Firm\_S_{it} + \alpha_8 ROE_{it} + \alpha_9 Lev\_R_{it} + \varepsilon_{it} \qquad (3)$$

　　　A complimentary test for the moderating effect of the crisis period (Cris_P) on both the relationship between ownership concentration and DDP and the relationship between CEO duality and DDP consists of using an interaction variable analysis. These interaction variables are the multiplication result between ownership concentration and the dummy Cris_P variable on the one hand, and CEO duality and the dummy Cris_P variable on the other, which equals 1 if the year considered is a crisis period and 0 otherwise. These interaction variables are supposed to capture the effect of both ownership concentration

and CEO duality on DDP only if the period concerned is characterized by a year of crisis. Thus, testing H1 and H2 consists of observing an insignificant positive association or negative and significant association between the interaction variables (Cris_P*Own_C) and (Cris_P*Board_D) and PDD (model 4).

$$DDP_{it} = \alpha_0 + \alpha_1 Cris\_P^*Own\_C_{it} + \alpha_2 Cris\_P^*Board\_D_{it} + \alpha_3 Own\_M_{it} + \alpha_4 Own\_F_{it} + \alpha_5 Board\_S_{it} + \alpha_6 Board\_W_{it} + \alpha_7 Firm\_S_{it} + \alpha_8 ROE_{it} + \alpha_9 Lev\_R_{it} + \varepsilon_{it} \quad (4)$$

Table 1 below provides full descriptions of all the different variables used in this study.

**Table 1.** Variable definitions and measurements.

| Label Variables | Definitions | Measurements | References |
|---|---|---|---|
| **Dependent variables** | | | |
| Div_PR | Dividend payout ratio | Dividendper share divided by earnings per share | Rahmawati et al. (2018); Murtaza et al. (2020); Endang et al. (2020); El Ammari (2021); Dissanayake and Dissabandara (2021). |
| Div_Y | Dividend yield | Dividend per share divided by market price per share | Kien and Chen (2020); Chertel et al. (2020); El Ammari (2021); Bataineh (2021); Thompson and Adasi Manu (2021). |
| **Independent variables** | | | |
| *Moderating variable* | | | |
| Cris_P | Crisis period | Dummy variable which is equal to 1 if the year considered is a crisis period (2010, 2011 and 2012 periods), and 0 otherwise. | Mehdi et al. (2017); Taleb and Ben Lahouel (2020). |
| *Test variables* | | | |
| Own_C | Ownership concentration | Herfindahl concentration index equal sum of the squared ownership shares by the five largest shareholders | Mossadak et al. (2016); Gonzalez et al. (2017); Krismiaji and Jati (2018); Chertel et al. (2020); El Ammari (2021). |
| Board_D | CEO Duality | Dichotomous variable which equals 1 when the CEO is also the board chairman, and 0 when the two roles are separated. | Shahid et al. (2016); Tahir et al. (2020c); Turrent et al. (2020); Ain et al. (2021); Dissanayake and Dissabandara (2021). |
| *Control variables* | | | |
| Own_M | Managerial ownership | Percentage of the shares held by top management (CEO) | Kulathunga and Azeez (2016); Krismiaji and Jati (2018); El Ammari (2021). |
| Own_F | Family ownership | Percentage of ownership owned by members of the same family. | Gana and El Ammari (2013); Reyna (2017); Taleb and Ben Lahouel (2020). |
| Board_S | Board size | Total number of directors present on the board | Abdelsalam et al. (2008); Boubaker and Nguyen (2012); Shahid et al. (2016); Tahir et al. (2020c); Ain et al. (2021); Dissanayake and Dissabandara (2021). |
| Board_W | Women's presence on boards | Dummy variable which is equal to 1 when there is at least one woman on the board of directors, and 0 otherwise. | Boubaker et al. (2014); Terjesen et al. (2016); Chen et al. (2017). |
| Firm_S | Firm size | Logarithm of total assets | Gana and El Ammari (2008, 2009); Shahid et al. (2016); Ain et al. (2021). |
| Lev_R | Leverage ratio | Ratio of the book value of total debt (short-and long-term) divided by the book value of total assets | Sindhu et al. (2016); Tahir et al. (2020c); Endang et al. (2020); Dissanayake and Dissabandara (2021); Zhao and Ng (2021). |
| ROE | Return on equity | Net income divided by shareholder's equity | Taleb and Ben Lahouel (2020); Ngatno et al. (2021); Jurado et al. (2021). |

### 3.3. Data Collection and Sample

To analyze the behavior of Tunisian companies in terms of DDP and its relationship with ownership structure and the board of directors' characteristics, companies listed on the TSE represent our total population. The data collected between 1996 and 2019 are taken from the financial statements of the selected companies, annual activity reports, and official bulletins of the TSE and the Financial Market Council. The ownership structure and



board of directors' data were collected manually from company annual reports and stock market guides, which can be downloaded from the TSE website and companies' official websites. For the period studied, the requested data was available for only 24 companies. For the other companies, financial and governance data is not available or is unreliable. Furthermore, in line with previous studies, we indicate that companies belonging to the financial sectors (e.g., banks, insurance, and leasing companies) were excluded from our sample due to their specific financial characteristics (in terms of the financial data, DDP, and governance systems) compared to their counterparts belonging to the non-financial sector. Accordingly, our final sample consists of 576 firm-year observations.

## 4. Empirical Results and Discussion

### 4.1. Descriptive Statistics

Table 2 provides an overview of descriptive statistics for all variables used in the study. Overall, the dividend payout ratio records an average of 20.90%. It indicates that the dividends distributed by a firm to the shareholders are still relatively low and are not optimal. Furthermore, the dividend yield has an average of 3.60% with a maximum of 17.40%. This value is slightly higher than that reported by Taleb and Ben Lahouel (2020), who find an average dividend yield of around 2.50% during the 2008–2019 period based on 44 firms studied in the Tunisian context. Our preliminary statistics also show that more than three-quarters of the firms in our sample have a fairly concentrated ownership structure, i.e., around 84.90%. It is indicated that there is a concentration of ownership by a very high percentage. The figures also indicate that about 80.40% of firms have a general manager who also chairs the board of directors. This is consistent with the findings of Wellalage and Locke (2011), who locate high duality in emerging countries. Nevertheless, this proportion is higher than those captured by Kim et al. (2009) in developed countries. This is not surprising in an emerging country like Tunisia, which is characterized by concentrated ownership where controlling shareholders tend to strengthen their control power by combining the management and supervisory functions.

**Table 2.** Descriptive statistics for the sample.

| Variables | Observations | Min | 1st Quartile | Mean | Median | 3rd Quartile | Max |
|---|---|---|---|---|---|---|---|
| Div_PR | 576 | 0 | 0.167 | 0.209 | 0.398 | 0.688 | 0.798 |
| Div_Y | 576 | 0 | 0.078 | 0.036 | 0.102 | 0.140 | 0.174 |
| Own_C | 576 | 0 | 0.406 | 0.849 | 0.664 | 0.892 | 1 |
| Board_D | 576 | 0 | 0.242 | 0.804 | 0.696 | 0.902 | 1 |
| Own_M | 576 | 0 | 0.384 | 0.605 | 0.504 | 0.960 | 1 |
| Own_F | 576 | 0 | 0.224 | 0.625 | 0.442 | 0.880 | 1 |
| Board_S | 576 | 3 | 3.032 | 7.750 | 7.026 | 12.430 | 13 |
| Board_W | 576 | 0 | 0.213 | 0.416 | 0.401 | 0.865 | 1 |
| Firm_S | 576 | 12.748 | 5.700 | 17.140 | 14.020 | 20.300 | 22.785 |
| Lev_R | 576 | 0.020 | 0.120 | 0.408 | 0.416 | 1.046 | 1.364 |
| ROE | 576 | −1.250 | 0.094 | 0.132 | 0.124 | 1.212 | 1.784 |

Notes: Div_PR: Dividend payout ratio; Div_Y: Dividend yield; Own_C: Ownership concentration; Board_D: CEO Duality; Cris_P: Crisis period; Own_M: Managerial ownership; Own_F: Family ownership; Board_S: Board size; Board_W: Women's presence on boards; Firm_S: Firm size; Lev_R: Leverage ratio; ROE: Return on equity.

On another level, Table 2 also indicates that the mean board size is approximately 7.75. The board size in Tunisia looks much smaller than the board size in the developed economies (e.g., the US mean board size is 11.45, Bhagat and Black (2002)). The leverage variable (Lev_R) reveals that the mean was 40.80%. It indicates that the needs of companies' funds are financed by debt which is relatively high. Furthermore, the results in Table 2 show that 41.60% of the companies studied have at least one woman on their board of directors, with a maximum of five female directors. Family businesses[1] represent approximately 62.50% of the population of companies studied.

### 4.2. Univariate Analysis

Table 3 presents the results of the univariate analysis while tracing the nature and intensity of the Pearson[2] and the Spearman correlation between the different variables of the model taken two by two as well as their degrees of significance. Several interesting relationships can be highlighted. First of all, the findings show that there is a strong significant negative relationship between the crisis period and both Div_PR and Div_Y with a Pearson correlation coefficient amounting to −0.385 and −0.256 respectively. This result provides preliminary support for H3. Besides, ownership concentration is positively and significantly correlated with both Div_PR and Div_Y with a Pearson correlation coefficient amounting to 0.247 and 0.232, respectively. Furthermore, the CEO duality is positively associated with Div_PR and Div_Y with a Pearson correlation coefficient of 0.428 and 0.108 respectively. Overall, these results provide preliminary support for our theoretical predictions. For the control variables, the firm and board size and ROE variables are positively correlated with Div_PR and Div_Y. However, only the variable that provides information on family property and leverage ratio is negatively related to Div_PR and Div_Y.

On another level, we notice the absence of a strong correlation between the different independent variables, which suggests the absence of a multicollinearity problem in our basic model. To further verify the absence of a multicollinearity problem, we conduct an additional test which allowed us to ensure that the VIF (variance inflation factor) values are sufficiently low (they vary between 1.06 and 4.73, which is largely below the critical value of 10). This confirms the absence of multicollinearity problems.

This table shows the Spearman and the Pearson correlations between the main variables used in the study and their degrees of significance. Spearman (Pearson) correlations appear above (below) the diagonal. The sample includes 576 firm-year observations over the 1996–2019 period.

### 4.3. Multivariate Analysis

To estimate our models with adequate methods, some tests are performed. Homogeneity (Hsiao) and Hausman tests indicate that there are individual fixed effects in our panel. In addition, the absence of endogeneity allows us to estimate our models using fixed-effect method[3].

Tables 4 and 5 present the results of multiple regressions specified in model 1. In model 1, the results show that the variable reflecting the degree of ownership concentration (Own_C) has a positive and significant effect on the dividend payout ratio (coefficient = 0.278; t = 2.880) and dividend yield (coefficient = 0.287; t = 3.820). This result confirms our initial hypothesis that: under concentrated ownership, Tunisian majority shareholders demand high levels of dividends to reduce agency conflicts. The idea is to mitigate the risks of mismanaging the excess cash available in the hands of managers (Mossadak et al. 2016). In addition, it can be assumed that increased oversight by majority shareholders could ensure that fewer company resources are wasted on poor-quality projects, implying that more cash flow can be paid out as dividends. These findings are in line with those of Abdullah et al. (2012) and Renneboog and Szilagyi (2020) in the German context. The latter argue that, in the context of concentrated ownership firms, high dividends payout moderates the concern of minority shareholders about the risk of expropriation of the firm's wealth by the management and therefore strengthens the firm's reputation in the financial market. Thus, consistent with Suwaidan and Khalaf (2020) in the Jordanian context, we suggest that ownership concentration appears to play an important role in the decisions of firms with concentrated ownership on DDP in the context of emerging countries.

**Table 3.** Spearman/Pearson Correlation Matrix. The Spearman correlations are above the diagonal and Pearson rank-correlation are below the diagonal.

| Variables | Div_PR | Div_Y | Own_C | Board_D | Cris | Own_M | Own_F | Board_S | Board_W | Firm_S | Lev_R | ROE |
|---|---|---|---|---|---|---|---|---|---|---|---|---|
| Div_PR | 1 | 0.258 *** | 0.364 * | 0.456 ** | −0.400 *** | −0.109 ns | −0.174 ns | 0.304 ** | 0.076 ns | 0.388 ** | −0.245 ns | 0.564 ** |
| Div_Y | 0.239 ** | 1 | 0.252 ** | 0.169 ** | −0.266 ** | −0.025 ns | −0.286 ns | 0.134 * | 0.082 ns | −0.068 ns | −0.221 ns | 0.298 *** |
| Own_C | 0.247 *** | 0.232 ** | 1 | 0.260 ** | −0.208 ** | 0.262 * | −0.299 ns | 0.062 ** | 0.293 ns | 0.321 ns | −0.198 ** | 0.036 ** |
| Board_D | 0.428 ** | 0.158 * | 0.243 *** | 1 | 0.221 ns | −0.289 ** | −0.060 * | 0.191 * | 0.220 * | 0.256 * | −0.040 ns | 0.412 ** |
| Cris | −0.385 *** | −0.256 ** | −0.197 ** | 0.215 ns | 1 | −0.080 ns | −0.026 * | −0.012 ** | 0.165 * | 0.098 ** | −0.330 ** | 0.110 ns |
| Own_M | −0.106 ns | −0.024 ns | 0.154 ** | 0.224 ** | −0.068 ns | 1 | 0.265 ns | 0.016 * | 0.178 ns | −0.404 *** | 0.192 ** | −0.287 * |
| Own_F | −0.162 ns | −0.168 ns | −0.253 ns | 0.024 ns | −0.008 * | 0.183 ns | 1 | −0.120 * | 0.332 * | −0.363 ns | 0.155 ns | −0.096 * |
| Board_S | 0.192 ** | 0.121 * | 0.027 ** | 0.173 * | −0.009 ** | 0.011 * | −0.040 * | 1 | 0.103 ns | 0.062 ns | 0.067 * | 0.024 ** |
| Board_W | 0.072 ns | 0.088 ns | 0.219 ns | 0.010 * | 0.018 ns | 0.126 ns | 0.182 * | 0.087 ns | 1 | 0.441 * | 0.136 ns | 0.230 ** |
| Firm_S | 0.336 ** | 0.074 * | 0.134 ns | 0.201 ** | 0.094 * | −0.201 ** | −0.302 ns | 0.033 ns | 0.402 * | 1 | −0.209 ns | 0.483 ** |
| Lev_R | −0.201 ns | −0.112 ns | −0.185 ** | −0.012 ns | −0.302 ** | 0.198 * | 0.078 ns | 0.014 * | 0.109 ns | −0.123 ns | 1 | 0.329 ** |
| ROE | 0.545 ** | 0.247 ** | 0.029 * | 0.303 ** | 0.104 ns | −0.259 * | −0.098 * | 0.012 ** | 0.213 ** | 0.442 ** | 0.304 ** | 1 |

Notes: Div_PR: Dividend payout ratio; Div_Y: Dividend yield; Own_C: Ownership concentration; Board_D: CEO Duality; Cris_P: Crisis period; Own_M: Managerial ownership; Own_F: Family ownership; Board_S: Board size; Board_W: Women's presence on boards; Firm_S: Firm size; ROE: Return on equity; Lev_R: Leverage ratio. *, **, and *** indicate significance at the 10%, 5%, and 1% level, respectively; ns: not significant.

**Table 4.** Multivariate regression analysis.

| | | | | | | | | | | |
|---|---|---|---|---|---|---|---|---|---|---|
| | **Dependant Variable: Div_PR** | | | | | | | | | |
| | **Model 1** **without Crisis** | | **Model 2** **with Crisis** | | **Model 3** **before Crisis** | | **Model 3** **during Crisis** | | **Model 3** **after Crisis** | |
| | Coefficient | *t-Statistic* | Coefficient | *t-Statistic* | Coefficient | *t-Statistic* | Coefficient | *t-Statistic* | Coefficient | *t-Statistic* |
| Intercept | −0.242 | −1.280 ns | −0.072 | −1.420 ns | −0.664 | −1.370 ns | −0.103 | −1.440 ns | −0.251 | −1.250 ns |
| *Own_C* | *0.278* | *2.880 ** | *−0.143* | *−1.920 * | *0.186* | *2.280 ** | *−0.059* | *−0.650 ns | *−0.224* | *−2.080 ** |
| *Board_D* | *0.245* | *3.450 *** | *0.068* | *1.110 ns | *0.169* | *2.560 ** | *0.002* | *1.480 ns | *0.056* | *1.620 ns |
| *Cris_P* | | | *−0.240* | *−3.470 *** | | | | | | |
| Own_M | −0.228 | −1.260 ns | −0.086 | −1.490 ns | 0.258 | 0.280 ns | −0.026 | −1.380 ns | 0.170 | 1.290 ns |
| Own_F | −0.147 | −1.740 * | −0.260 | −2.010 ** | −0.256 | −0.290 ns | −0.066 | −1.810 * | −0.014 | −1.830 * |
| Board_S | 0.095 | 2.350 ** | 0.102 | 1.070 ns | 0.074 | 1.850 * | 0.102 | 1.730 * | 0.064 | 1.700 * |
| Board_W | 0.126 | 1.520 ns | −0.019 | −0.780 ns | 0.312 | 0.150 ns | −0.186 ns | −1.210 ns | −0.171 | −0.780 ns |
| Firm_S | 0.028 | 2.440 ** | 0.008 | 1.830 * | 0.093 | 0.540 ns | 0.212 | 2.840 ** | 0.086 | 2.990 ** |
| ROE | 0.096 | 3.840 *** | 0.085 | 1.790 * | 0.077 | 3.770 *** | 0.098 | 2.870 ** | 0.080 | 2.770 ** |
| Lev_R | −0.134 | −1.420 ns | −0.018 | −0.180 ns | −0.060 | −0.080 ns | −0.158 | −1.320 ns | −0.214 | −1.460 |
| F (p-value) | 16.390 *** (0.000) | | 23.540 *** (0.000) | | 16.730 *** (0.000) | | 10.420 *** (0.000) | | 18.000 *** (0.000) | |
| Adj. R² (%) | 32.840 | | 39.160 | | 25.560 | | 24.680 | | 26.400 | |
| Max VIF | 3.240 | | 3.320 | | 3.140 | | 1.570 | | 2.340 | |
| Number of observations | 576 | | 576 | | 336 | | 72 | | 168 | |

Notes: Div_PR: Dividend payout ratio; Own_C: Ownership concentration; Board_D: CEO Duality; Cris_P: Crisis period; Own_M: Managerial ownership; Own_F: Family ownership; Board_S: Board size; Board_W: Women's presence on boards; Firm_S: Firm size; ROE: Return on equity; Lev_R: Leverage ratio. *, **, and *** indicate significance at the 10%, 5%, and 1% level, respectively; ns: not significant; italic style is used to highlight the variables of interest in the models.

**Table 5.** Multivariate regression analysis.

| | Dependant Variable: Div_Y | | | | | | | | | |
|---|---|---|---|---|---|---|---|---|---|---|
| | Model 1 without Crisis | | Model 2 with Crisis | | Model 3 before Crisis | | Model 3 during Crisis | | Model 3 after Crisis | |
| | Coefficient | *t-Statistic* | Coefficient | *t-Statistic* | Coefficient | *t-Statistic* | Coefficient | *t-Statistic* | Coefficient | *t-Statistic* |
| Intercept | −0.042 | −1.370 $^{ns}$ | −0.028 | −1.520 $^{ns}$ | −0.607 | −1.130 $^{ns}$ | −0.004 | −1.450 $^{ns}$ | −0.520 | −1.020 $^{ns}$ |
| *Own_C* | *0.287* | *3.820 \*\*\** | *−0.224* | *−1.970 \** | *0.087* | *2. 720 \*\** | *−0. 609* | *−0.580 $^{ns}$* | *−0.029* | *−2.024 \*\** |
| *Board_D* | *0.254* | *1. 940 \** | *0.068* | *1.110 $^{ns}$* | *0.096* | *1.680 \** | *0.032* | *1.080 $^{ns}$* | *0.064* | *1.250 $^{ns}$* |
| *Cris_P* | | | *−0.430* | *−2.750 \*\** | | | | | | |
| Own_M | −0.272 | −1.602 $^{ns}$ | −0.066 | −1.520 $^{ns}$ | 0.028 | 0.830 $^{ns}$ | −0.052 | −1.490 $^{ns}$ | 0.069 | 1.380 $^{ns}$ |
| Own_F | −0.044 | −1.920 * | −0.009 | −2.320 ** | −0.362 | −0.710 $^{ns}$ | −0.027 | −2.980 ** | −0.141 | −1.740 * |
| Board_S | 0.058 | 1.690 * | 0.002 | 1.030 $^{ns}$ | 0.017 | 1.900 * | 0.021 | 2.370 ** | 0.007 | 1.810 * |
| Board_W | 0.260 | 1.430 $^{ns}$ | −0.001 | −0.270 $^{ns}$ | 0.613 | 0.240 $^{ns}$ | −0.195 $^{ns}$ | −0.910 $^{ns}$ | 0.011 | 0. 870 $^{ns}$ |
| Firm_S | 0.048 | 2.840 ** | 0.072 | 1.160 $^{ns}$ | 0.032 | 0.760 $^{ns}$ | 0.122 | 2.760 ** | 0.009 | 2.590 ** |
| ROE | 0.669 | 3.120 *** | 0.068 | 2.690 ** | 0.876 | 2.670 ** | 0.679 | 1.770 * | 0.086 | 2.630 ** |
| Lev_R | −0.013 | −0.210 $^{ns}$ | −0.109 | −0.470 $^{ns}$ | −0.050 | −1.230 $^{ns}$ | −0.385 | −1.130 $^{ns}$ | −0.123 | −0.270 |
| F (p-value) | 18.060 *** (0.000) | | 20.350 *** (0.000) | | 10.900 *** (0.000) | | 8.240 *** (0.000) | | 12.270 *** (0.000) | |
| Adj. R$^2$ (%) | 28.240 | | 36.610 | | 22.360 | | 20.860 | | 28.200 | |
| Max VIF | 3.130 | | 3.430 | | 3.570 | | 1.860 | | 2.470 | |
| Number of observations | 576 | | 576 | | 336 | | 72 | | 168 | |

Notes: Div_Y: Dividend yield; Own_C: Ownership concentration; Board_D: CEO Duality; Cris_P: Crisis period; Own_M: Managerial ownership; Own_F: Family ownership; Board_S: Board size; Board_W: Women's presence on boards; Firm_S: Firm size; ROE: Return on equity; Lev_R: Leverage ratio. *, **, and *** indicate significance at the 10%, 5%, and 1% level, respectively; $^{ns}$: not significant; italic style is used to highlight the variables of interest in the models.

Another interesting result emerges from our model 1. The findings indicate that when the Chief Executive Officer (CEO) is at the same time chair of the board (COB), Tunisian firms are inclined to pay out more dividends than when the two functions are held by two different persons. This relationship is valued by the Div_PR and Div_Y variables. Indeed, the coefficient of the Board_D variable is positive and statistically highly significant at the 1% threshold (coefficient = 0.245; t = 3.450). This impact is much less intense on the second variable (coefficient = 0.254; t = 1.940). This finding is not aligned with hypothesis H2, which predicts a negative impact of CEO duality on the dividend payment. This situation seems likely to reinforce the discretionary power of the controlling shareholders, since the CEO is often a member of the controlling family, or chosen by the controlling entity. Rather, this result corroborates the findings of Taleb and Ben Lahouel (2020) in the Tunisian context and Suwaidan and Khalaf (2020) in the Jordanian context, which suggests that firms with a CEO who accumulates both management and control functions tend to pursue a generous DDP. Furthermore, Mehdi et al. (2017) accounted for this positive relationship by the idea that CEO duality cannot be regarded as an efficient means to soothe expropriation risk in emerging markets (such as the Gulf Cooperation Council (GCC) and East Asia). As a result, shareholders demand greater dividend payouts to solve free cash flow issues. Such a viewpoint validates the results of Bradford et al. (2013) who assert that in developed countries CEO duality is not an adequate control mechanism.

Regarding the remaining control variables, the results obtained detect a positive and significant relationship between the board size and Tunisian companies' DDP. This relationship is assessed by the two variables dividend payout ratio (coefficient = 0.095; t = 2.350) and dividend yield (coefficient = 0.058; t = 1.690), respectively at 5% and 10% thresholds. This conclusion is consistent with those of Hermalin and Weisbach (2003); Ntim et al. (2015) and Taleb and Ben Lahouel (2020) who stipulate that the complexity of large boards can be ineffective, due to slow decision making as well as difficulty in communication and coordination between stakeholders. To this end, the level of dividend distribution should be higher in order to compensate for poor governance practices adopted by members of the management team. Likewise, firm size and ROE are positively and statistically significantly associated with DDP, while family ownership is negatively related to the same variable ((coefficient = −0.448; t = −1.740) and (coefficient = −0.044; t = −1.920) for Div_PR and Div_Y variable respectively). For the other variables—managerial ownership, and leverage ratio (Women's presence on boards)—they show positive (negative) coefficients but are not statistically significant. This indicates that neither the managerial ownership nor the leverage ratio, as well as women's presence on boards, seems to affect DDP.

Controlling for multicollinearity, the reported variance inflation factors (VIFs) suggest that model 1 in Tables 4 and 5 does not suffer from such a problem as the maximum VIF accounts for 3.240 and 3.130 respectively[4]. The overall explanatory power of the model is significantly high ((F = 16.390; $p < 0.000$ and the adjusted-R2 accounts for 32.840%) and (F = 18.060; $p < 0.000$ and the adjusted-R2 accounts for 28.240%) in Tables 4 and 5 respectively)).

In model 2, the variable Cris_P is added to study its impact on the DDP and test, at the same time, how the degree of both ownership concentration and CEO duality can influence the DDP in the presence of the crisis period. From the results obtained in model 2 (Tables 4 and 5), we show that the dividend payout ratio and dividend yield are significantly and negatively impacted in crisis period ((coefficient = −0.240; t = −3.470) and (coefficient = −0.430; t = −2.750) for Div_PR and Div_Y variable respectively). This is supported by the findings of Mehdi et al. (2017) who focused on the effect of the 2008 financial crisis on the dividends payout decision of the Gulf Cooperation Council (GCC) and East Asia countries. Such a negative correlation is due to the fact that board members become more risk-diffident in periods of crisis. Additionally, this negative correlation shows that crisis periods tremendously cripple Tunisian firms' dividend payouts. Indeed, in times of financial, economic, or political unrest, it is difficult to increase external funds, especially bank financing. Thus, firms opt to keep a great share of their profits tothe

detriment of dividend distribution. During these difficult periods, firms want to enhance their systems of governance and safeguard their shareholders as much as possible by internally reinvesting their profits.

The overall explanatory power of the model becomes stronger ((F = 23.540; $p$ < 0.000 and the adjusted-R2 accounts for 39.160%) and (F = 20.350; $p$< 0.000 and the adjusted-R2 accounts for 36.610%) in Tables 4 and 5 respectively). For model 2, the adjusted-R2 in Table 4 (Table 5) witnesses a significant increase, moving from 32.840% to 39.160% (28.240 to 36.610), implying that the Cris_P represents the most important variable in explaining DDP. This result confirms the strong univariate correlation between the Cris_P and DDP.

However, an interesting result emerges when we compare model 1 and model 2. Taking the three explanatory variables (Cris_P, Own_C, and Board_D) together in the same model, the results show that the relationship between ownership concentration and DDP becomes negative and significant ((coefficient = −0.143; t = −1.920) and (coefficient = −0.224; t = −1.970) for Div_PR and Div_Y variable respectively) as compared to model 1, while the CEO duality becomes insignificantly associated with the same variable. This implies that during crisis periods, Tunisian firms with CEO duality and a high ownership concentration pay lower cash dividends. The results thus prove that a crisis period would affect management policies adopted by the Tunisian majority shareholders and the management team, such as dividend policy. Accordingly, Cris_P on the one hand, and both ownership concentration and CEO duality on the other, represent two concurrent forces influencing DDP with a prevailing negative effect for Cris_P as it transforms the positive effect of ownership concentration into a negative effect and attenuates, at the same time, the positive effect of CEO duality on DDP to make it insignificant.

To test how Cris_P may affect the relationship between both ownership concentration and DDP, as well as between CEO duality and DDP, the overall sample is sub-grouped into three periods namely: period before crisis, period during crisis, and period after crisis. In model 3, the results obtained demonstrate that the significant positive association between the ownership concentration and the DDP observed in model 1 remains significant for the period before the crisis ((coefficient = 0.186; t = 2.280) and (coefficient = 0.087; t = 2.720) for Div_PR and Div_Y variable respectively), while it becomes insignificant during the period of crisis to transform into a significant negative effect after the crisis period. Additionally, the results reveal that the significant positive relationship between CEO duality and DDP observed in model 1 remains significant for the pre-crisis period ((coefficient = 0.169; t = 2.560) and (coefficient = 0.096; t = 1.680) for Div_PR and Div_Y variable respectively), while it becomes insignificant during and after the crisis period.

Checking for the multicollinearity problem, models 3 and 4 in Table 4 (Table 5) do not suffer from this problem as all maximum VIFs account for 3.140, 1.570, 2.340, and 2.560 (3.570, 1.860, 2.470, and 3.640) in model 3 before crisis, model 3 during crisis, model 3 after crisis, and model 4, respectively.

Overall, the findings provide evidence that the political instability and financial crisis that occurred between 2010 and 2012 have weakened the positive association between ownership concentration, CEO duality, and DDP in the Tunisian setting. This implies that major shareholders in the Tunisian listed companies opt for a conservative dividend policy during political instability periods. This also implies that they adopt a strategy of less cash distribution since political instability is generally characterized by a high level of financial and operating risks (Khlif et al. 2019).

### 4.4. Alternative Regressions for Model 4

This sub-section is devoted to undertaking further tests concerning model 4 as shown in Tables 6 and 7. The first model includes the interaction variables (Cris_P*Own_C) and (Cris_P*Board_D), ownership concentration, CEO duality, and the Cris_P. The reported results show that neither ownership concentration, as well as CEO duality nor the interaction variables, have a significant effect on DDP. By contrast, the Cris_P conserves its strong negative impact on DDP ((coefficient = −0.653; t = −3.640) and (coefficient = −0.330;

t = −2.870) for Div_PR and Div_Y variable respectively). These findings confirm that the Cris_P has a prevailing negative effect on DDP over both ownership concentration and CEO duality. At the level of the second alternative model, the variable Cris_P is eliminated. In this regression, both ownership concentration and CEO duality have a positive impact on DDP, while the interaction variables between ownership concentration and the Cris_P on the one hand and between CEO duality and Cris_P on the other have an opposite effect on the same variable. These findings confirm that Cris_P represents a concurrent force for ownership concentration and CEO duality and mitigates its adverse impact on DDP.

**Table 6.** Alternative regression for model 4.

| | Dependant Variable: Div_PR | | | |
|---|---|---|---|---|
| | **Model 4** **with Own_C, Board_D and Cris_P** | | **Model 4** **with Own_C and Board_D** | |
| | **Coefficient** | ***t*-Statistic** | **Coefficient** | ***t*-Statistic** |
| Intercept | −0.223 | −1.200 ns | −0.712 | −1.330 ns |
| *Own_C* | *0.127* | *1.180 ns* | *0.334* | *2.270 *** |
| *Board_D* | *0.015* | *1.040 ns* | *0.276* | *1.990 ** |
| *Cris_P* | *−0.653* | *−3.640 **** | | |
| *Cris_P* Own_C* | *−0.026* | *−0.040* | *−0.830* | *−3.530 **** |
| *Cris_P*Board_D* | *−0.021* | *−0.630* | *−0.026* | *−4.140 **** |
| Own_M | −0.028 | −0.160 ns | −0.621 | −1.220 ns |
| Own_F | −0.247 | −1.660 * | −0.840 | −2.810 ** |
| Board_S | 0.095 | 2.350 ** | 0.203 | 1.030 ns |
| Board_W | 0.126 | 1.520 ns | 0.027 | 0.090 ns |
| Firm_S | 0.028 | 2.440 ** | 0.418 | 1.920 * |
| ROE | 0.096 | 2.840 ** | 0.008 | 1.690 * |
| Lev_R | −0.134 | −1.420 ns | −0.010 | −0.110 ns |
| F (*p*-value) | 21.360 *** (0.000) | | 18.740 *** (0.000) | |
| Adj. $R^2$ (%) | 38.180 | | 35.450 | |
| Max VIF | 3.720 | | 3.260 | |
| Number of observations | 576 | | 576 | |

Notes: Div_PR: Dividend payout ratio; Own_C: Ownership concentration; Board_D: CEO Duality; Cris_P: Crisis period; Own_M: Managerial ownership; Own_F: Family ownership; Board_S: Board size; Board_W: Women's presence on boards; Firm_S: Firm size; ROE: Return on equity; Lev_R: Leverage ratio. *, **, and *** indicate significance at the 10%, 5%, and 1% thresholds, respectively; ns: not significant; italic style is used to highlight the variables of interest in the models.

**Table 7.** Alternative regression for model 4.

| | Dependant Variable: Div_Y | | | |
|---|---|---|---|---|
| | **Model 4** **with Own_C, Board_D and Cris_P** | | **Model 4** **with Own_C and Board_D** | |
| | **Coefficient** | ***t*-Statistic** | **Coefficient** | ***t*-Statistic** |
| Intercept. | −1.032 | −1.810 ns | −0.072 | −1.420 ns |
| *Own_C* | *0.308* | *0.770 ns* | *0.187* | *1.940 ** |
| *Board_D* | *0.225* | *0.640 ns* | *0.264* | *1.080 ns* |
| *Cris_P* | *−0.330* | *−2.870 *** | | |
| *Cris_P* Own_C* | *−0.006* | *−1. 310 ns* | *−0.060* | *−3. 910 **** |
| *Cris_P*Board_D* | *−0.012* | *−0.110 ns* | *−0.062* | *−4.110 **** |
| Own_M | −0.527 | −1.060 ns | −0.086 | −1.490 ns |
| Own_F | −0.384 | −1.850 * | −0.560 | −2.010 ** |
| Board_S | 0.076 | 2.260 ** | 0.102 | 1.070 ns |
| Board_W | 0.106 | 0.620 ns | 0.019 | 0.180 ns |
| Firm_S | 0.003 | 2.540 ** | 0.048 | 1.830 * |
| ROE | 0.069 | 3.630 *** | 0.085 | 1.790 * |
| Lev_R | −0.243 | −0.220 ns | −0.018 | −0.180 ns |
| F (p-value) | 23.240 *** (0.000) | | 20.00 *** (0.000) | |
| Adj. $R^2$ (%) | 37.520 | | 28.830 | |
| Max VIF | 3.810 | | 3.340 | |
| Number of observations | 576 | | 576 | |

Notes: Div_Y: Dividend yield; Own_C: Ownership concentration; Board_D: CEO Duality; Cris_P: Crisis period; Own_M: Managerial ownership; Own_F: Family ownership; Board_S: Board size; Board_W: Women's presence on boards; Firm_S: Firm size; ROE: Return on equity; Lev_R: Leverage ratio. *, **, and *** indicate significance at the 10%, 5%, and 1% level, respectively; ns: not significant; italic style is used to highlight the variables of interest in the models.

## 5. Conclusions and Implications

The issue of the DDP has raised many questions and remains a real subject of debate. The results of various researches on this point to date remain inconclusive. Our research follows this line by proposing to examine the effect of ownership concentration and CEO duality on DDP for a sample composed of 576 firm-year observations from 24 Tunisian listed firms over the 24-year period (1996 to 2019). An interesting contribution of our study is to test whether the impact of ownership structure and CEO duality change during a crisis period. Moreover, another major contribution is to test, like Zulfikar et al. (2020) and Ngatno et al. (2021), using an interaction variable, the moderating effect of the crisis period on the association between both the degree of ownership concentration and DDP on the one hand, and between the CEO duality and DDP on the other.

Our results provide strong evidence of the important role of ownership concentration and CEO duality in accounting for DDP mainly in a crisis period. Thanks to our study, the dividend policy concept will be better grasped by actual and potential investors. Hence, they know the correlation between payout implications and management decisions and the impact of this correlation on their wealth. Their investment decisions, now, can be made based on correct and empirically tested data. Additionally, the results show that the crisis period is negatively associated with the DDP. Focusing on the role of the crisis period in studying the DDP, we show that the effect of ownership concentration and CEO duality on DDP varies depending on the period studied before, during, and after the crisis in emerging countries. The main implications are that during and after the crisis period, companies with CEO duality and ownership concentration are more risk-averse and therefore prefer not to pay dividends during the crisis period. The results obtained also show that the period of crisis weakens the positive association between both ownership concentration and CEO duality on DDP. Moreover, the crisis period on the one hand and both ownership concentration and CEO duality on the other represent two competing forces influencing DDP in a country with a predominantly negative impact for the period of crisis. Our findings imply that major shareholders in Tunisian listed companies opt for a conservative dividend policy during political instability periods and adopt a strategy of less cash distribution since political instability is generally characterized by a high level of financial and operating risks (Khlif et al. 2019).

Theoretical and practical implications can be deduced from our findings. Theoretically, this study brings theoretical contributions to the field of financial research by exploring this field in Tunisia and more generally in an emerging economy. The findings of this study contribute also to existing theories and may be used for future researchers to conduct further analysis and to draw comparisons of the results from other studies on the topic, in different geographical, economic, and political settings. Thus, the present paper is addressed not only to a Tunisian or a global audience within the research field of finance but also to firm managers and stakeholders. Furthermore, it contributes novel insights in to the impact of effective corporate governance mechanisms on the firm's dividend payout decisions.

In practice, this research paper is useful for Tunisian listed companies' board of directors to gain a better understanding of the influential factors of dividend policy. Besides, our results are very useful, particularly for managers and shareholders in order to rationalize and adjust their dividend payout strategies during a crises period. Indeed, having an efficient dividend scheme during periods of crisis allows the company to be more competitive and maximizes shareholders' profits. Subsequently, since ownership structure and the board of directors' characteristics are considered to be factors that disrupt the dividend decision, our findings can also be utilized by managers and financial analysts in the field of economics and finance to make knowledgeable and effective decisions in order to improve the firm's performance. Furthermore, these findings appear to be relevant given that in recent years investors have shown a growing interest in ownership structure and board of directors' characteristics (Graham et al. 2020; Suwaidan and Khalaf 2020).

However, like any other study, this research work has certain limitations. First, the limited number of non-financial companies listed on the TSE is the main limit. Indeed, the small size of the sample compels us not to consider and isolate the specificities of each sector and to work in a homogeneous environment. Furthermore, some other control variables may be included in the model; which implies that the findings may suffer from omitted variables bias. Finally, because of data unavailability, we were not able to conduct additional tests using alternative DDP proxies.

Concerning future avenues of research, researchers suggest examining other dimensions such as female directors serving on the board, director remuneration, audit, and risk as moderators on the relationship between ownership structure and CEO duality on DDP. They also might consider financial firms, which represent about half of the total listed companies on the TSE, and are known for the entrenchment of their politically connected directors. Lastly, because this study is limited to an emerging economy, namely the Tunisian market, researchers propose expanding the study sample internationally to determine whether country-specific factors (characterized by different cultures and regulations) may alter the conclusions reached.

**Funding:** This research received no external funding.

**Institutional Review Board Statement:** Not applicable.

**Informed Consent Statement:** Not applicable.

**Data Availability Statement:** The data presented in this study are available on request from the corresponding author. The data are not publicly available due to privacy.

**Acknowledgments:** The author wants to thank the three anonymous referees for their thoughtful comments and suggestions, which have dramatically contributed to the improvement of this paper.

**Conflicts of Interest:** The author declares no conflict of interest.

## Notes

[1] According to Gana and El Ammari (2013), a firm is considered family-owned only when shareholders of the same family hold at least 20 percent of the total shares.

[2] According to Bryman and Cramer (1997), Pearson's correlation between the independent variables is not reflected as a problem unless it is higher than 0.80, because independent variables with coefficients greater than 0.80 are supposed to show multicollinearity.

[3] The results are available from the author upon request.

[4] According to the rule of thumb, multicollinearity is viewed as a serious problem when the VIF exceeds 10 (Neter et al. 1989).

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
