# Peer review of "Do CEO Duality and Ownership Concentration Impact Dividend Policy in Emerging Markets? The Moderating Effect of Crises Period"

_ijfs, doi:10.3390/ijfs9040062_

Round 1

Reviewer 1 Report

I find the topic proposed in the paper interesting, although the work needs reassessments and adjustments to be publishable.

Title: Instead of Emerging Market I suggest Emerging Markets, this way the study would have applicability to other similar countries. Also, instead of "The moderate effect of Crisis period" I suggest "The moderate effect of crises"

Introduction:

  • It is too long, particularly in relation to Literature review. Many parts of the Introduction need to be moved to the Literature review section, because this is what they are, a review of literature. The Introduction should be limited to setting the research problem, explaining the rationale behind the study, discussing the originality of the paper, and presenting the structure of the paper.
  • The authors state that "By contrast, in emerging markets, this shareholder protection is not as effective as in their developed counterparts. In this sense, Abdelsalam et al. (2008) conclude that  the level of dividends is higher in emerging than in developed markets." I find this phrase contradictory and the authors need to explain this better. Moreover, what does "higher" mean here? In terms of what? Absolute amount? Percentage of distributed profits?
  • The authors affirm that "In this country, the companies are precisely characterized by extensively concentrated ownership and weak corporate governance compared with those in other developed countries", but no other information is provided. Please provide more information on this situation, so that readers that are not familiar with Tunisia understand the research. In this context, I advise the authors to discuss more the corporate governance issues in emerging countries and the contrast against developed countries, it would give more strength to their research.
  • I also have an observation on the two factors that are the basis of this study, CEO duality and ownership concentration. I expected the authors to relate them as a rationale of the research problem, but no such relation is provided. If this is not happening, then I do not understand why they are part of the same research and paper, and not examined in different papers.
  • Overall, I find the Introduction long (as mentioned above) but also rather disorganized, with ideas being repeated in various paragraphs. This does not offer a good impression to any reader.

Literature review: Once the Introduction is over, this section is nothing else but a presentation of the same ideas. I found nothing new here. That is why I think the authors need to reorganize these sections.

Hypotheses development:

  • This section doubles the Literature review section. I think this is not acceptable, the hypotheses should be developed based on what is presented in the literature review section, otherwise, a rather big part of the paper (1/3 of text) is basically a review of the literature. 
  • What does H1 mean? What does it mean that there is a significant association between ownership concentration and DDP in Tunisia? In what sense do the authors expect that ownership concentration acts? And what do the authors mean by DDP here? They need to provide a characteristic of DDP that is measurable, otherwise it cannot be tested. I have a similar comment for H2. For H3, the authors mention a negative connection to DDP, but DDP is not measurable.

Research design:

  • Which is the data frequency and what source of data have the authors used? In the model - equation (1) CEO duality and ownership concentration are moderating variables and the crisis is test variables. Based on what the authors mentioned until here, I think it should be the other way around.
  • It is not clear how the variables are defined and calculated; for example, how is the Herfindahl measure determined?
  • Why is Family ownership introduced? I believe this is not necessary, as this type of ownership is not of concern here and it puzzles the results. I also draw the attention to authors that ROE already includes financial leverage, therefore introducing both in the panel regression doubles the influence of leverage.

Results:

  • Overall, I find the results discussed rather simply, and the sample of firms included in the study is not presented: number of employees, turnover, evolutions over time are a must.
  • "It indicates that the dividends distributed by a firm to the shareholders are still relatively low and are not optimal." What does optimal mean here? 
  • The authors discuss board size in Tunisia and state that it is lower than in other countries. But how do the size of Tunisian firms and the extent of their operations compare against other countries?
  • Which method has been used to estimate the parameters in the panels and why?
  • "firms with CEO duality and a high ownership concentration pay lower cash dividends". Isn't this normal during a crisis period, for firms to pay less dividends?
  • The authors divide the period under scrutiny in three sub-periods, but they do not say which are these periods.
  • The Results section is very difficult to follow, I suggest the authors to reorganize it and to make it clearer in terms of findings and their meaning. The tables should be connected to the specified models in equations - as a note to the tables, the authors can mention the equations the results are based upon. Also, there are big tables one after the other, this creates a deplorable impression on the manuscript.

Conclusions: The authors refer mainly to the use of their research for Tunisia, but in order for the paper to be interesting for a wider audience they need to discuss the results and their relevance for other countries and companies. 

Author Response

Dear Editor,

I would like to thank you and the reviewers for your helpful comments and suggestions that were instrumental in dramatically improving our work. Below please find the detailed responses for each comment. All new revisions in the revised version of the manuscript are highlighted in yellow. Also, please consider the latest version of our manuscript as we have made numerous changes following the reviewers’ suggestions. We hope that these revisions meet you’re your expectations and those of the referees. 

Should you have any question or additional suggestions, please do not hesitate to contact me.

Best regards 

Dr. Anis El Ammari

Manuscript ID: ijfs-1374877

Reviewer 1 Comments:

I find the topic proposed in the paper interesting, although the work needs reassessments and adjustments to be publishable:

1/ Title: Instead of Emerging Market I suggest Emerging Markets, this way the study would have applicability to other similar countries. Also, instead of "The moderate effect of Crisis period" I suggest "The moderate effect of crises". 

Response:

Thank you very much for your suggestion. The title of the article has been modified according to the recommendations requested. It is now; “Do CEO Duality and Ownership Concentration Impact Dividend Policy in Emerging Markets? The Moderating effect of crises”

2/Introduction

It is too long, particularly in relation to Literature review. Many parts of the Introduction need to be moved to the Literature review section, because this is what they are, a review of literature. The Introduction should be limited to setting the research problem, explaining the rationale behind the study, discussing the originality of the paper, and presenting the structure of the paper.

The authors state that "By contrast, in emerging markets, this shareholder protection is not as effective as in their developed counterparts. In this sense, Abdelsalam et al. (2008) conclude that the level of dividends is higher in emerging than in developed markets." I find this phrase contradictory and the authors need to explain this better. Moreover, what does "higher" mean here? In terms of what? Absolute amount? Percentage of distributed profits?

The authors affirm that "In this country, the companies are precisely characterized by extensively concentrated ownership and weak corporate governance compared with those in other developed countries", but no other information is provided. Please provide more information on this situation, so that readers that are not familiar with Tunisia understand the research. In this context, I advise the authors to discuss more the corporate governance issues in emerging countries and the contrast against developed countries, it would give more strength to their research.

I also have an observation on the two factors that are the basis of this study, CEO duality and ownership concentration. I expected the authors to relate them as a rationale of the research problem, but no such relation is provided. If this is not happening, then I do not understand why they are part of the same research and paper, and not examined in different papers.

Overall, I find the Introduction long (as mentioned above) but also rather disorganized, with ideas being repeated in various paragraphs. This does not offer a good impression to any reader.

Response:

Many thanks for your detailed and insightful recommendations. As suggested, the introduction has now been better organized and more polished by reducing the content to the research problem, the rationale underlying the study, discussion of originality of the paper, and presenting the structure of the paper.

3/ Hypotheses development:

This section doubles the Literature review section. I think this is not acceptable, the hypotheses should be developed based on what is presented in the literature review section, otherwise, a rather big part of the paper (1/3 of text) is basically a review of the literature. 

What does H1 mean? What does it mean that there is a significant association between ownership concentration and DDP in Tunisia? In what sense do the authors expect that ownership concentration acts? And what do the authors mean by DDP here? They need to provide a characteristic of DDP that is measurable, otherwise it cannot be tested. I have a similar comment for H2. For H3, the authors mention a negative connection to DDP, but DDP is not measurable.

Response:

Thank you very much for your comments

-    In order to avoid any redundancy, the literature review section has now been deleted. We keep only the section on hypotheses development.

-    The rationale underlying the relationship between the ownership concentration and DDP in Tunisia is not directional as there are no conclusive arguments in favor of one particular direction. From an empirical side, there are some authors who show a positive association; others suggest a negative association, whereas many also show no association.

-   The dividend distribution policy (DDP) variables are measurable. We use two different dependent variables to define the DDP, namely, the dividend payout ratio (Div_PR) and the dividend yield (Div_Y).

4/ Research design:

Which is the data frequency and what source of data have the authors used? In the model - equation (1) CEO duality and ownership concentration are moderating variables and the crisis is test variables. Based on what the authors mentioned until here, I think it should be the other way around.

It is not clear how the variables are defined and calculated; for example, how is the Herfindahl measure determined?

Why is Family ownership introduced? I believe this is not necessary, as this type of ownership is not of concern here and it puzzles the results. I also draw the attention to authors that ROE already includes financial leverage, therefore introducing both in the panel regression doubles the influence of leverage.

Response:

Thank you very much for your thoughtful and constructive comments

-   The data used are annual data for the period 1996-2019. The data sources are financial statements of sample firms, annual reports, and official bulletins of the TSE and of the Financial Market Council. The ownership structure and board of directors’ data were collected manually from company annual reports and stock market guides, which can be downloaded from the TSE website and companies’ official websites.

-   We agree with the viewpoint of the reviewer that there is a mistake regarding the naming of the test variables and the moderating variable. To this end, we have made the required modification concerning the naming of these two variables (moderating variable: Crisis period; test variables: CEO duality and ownership concentration). 

-  The ownership concentration indicates the ownership level of major shareholders. We measure the concentration by the Herfindhal index. It is one of the most widely used measures in the literature (Harada and pascal, 2011; Pablo and Gonzalez, 2010). The Herfindhal index measures the concentration of market shares. In governance research, the Herfindhal index is used to calculate the ownership concentration level of company (Demsetz and Lehn, 1985). The index is equal to sum of the squared ownership shares by the five largest shareholders. This measure has been used by many authors such as Mossadak et al. (2016), Gonzalez et al. (2017), and Krismiaji and Jati (2018).

-  According to financial literature, family ownership influences dividend distribution policy. In fact, we control for the influence of family ownership on the dividend distribution policy since most of the study population is mainly controlled through a family. Indeed, the research on the dividends distribution of family firms shows that they distribute less than non-family. If the shareholder is a family, the control of the directors is better reinforced (Demsetz and Lehn, 1985; Gilson and Gordon, 2003) due to their better knowledge of the firms (Anderson and Reeb, 2003) and their long-term investment horizon. Similarly, in the same vein, several authors such as Ali et al. (2007) show that family ownership has a negative effect on distribution of dividends due to lower overall agency cost.

-   The use of the two variables (ROE and financial leverage) in the same model is justified by prior literature and the absence of multicollinearity. In fact, i) there is no multicollinearity problem between these two variables and ii) many authors such as Tahir and Mushtaq (2016); Rahmanet al. (2020); and Taleb and Ben Lahouel (2020), among others, have used these two variables in the same model.

5/ Results

Overall, I find the results discussed rather simply, and the sample of firms included in the study is not presented: number of employees, turnover, evolutions over time are a must.

"It indicates that the dividends distributed by a firm to the shareholders are still relatively low and are not optimal." What does optimal mean here? 

The authors discuss board size in Tunisia and state that it is lower than in other countries. But how do the size of Tunisian firms and the extent of their operations compare against other countries?

Which method has been used to estimate the parameters in the panels and why?

"firms with CEO duality and a high ownership concentration pay lower cash dividends". Isn't this normal during a crisis period, for firms to pay less dividends?

The authors divide the period under scrutiny in three sub-periods, but they do not say which are these periods.

The Results section is very difficult to follow, I suggest the authors to reorganize it and to make it clearer in terms of findings and their meaning. The tables should be connected to the specified models in equations - as a note to the tables, the authors can mention the equations the results are based upon. Also, there are big tables one after the other, this creates a deplorable impression on the manuscript.

Response:

We would like to thank you again for your insightful and constructive comments. 

-    We considered that the dividend payout rate is not optimal because descriptive statistics (Table 2) show that the average dividend payout ratio (0.209 and 0.036 for Div_PR and Div_Y variable, respectively) is very low compared to the maximum dividend payout ratio (0.798 and 0.174 for Div_PR and Div_Y variable, respectively).

-  "Firms with CEO duality and a high ownership concentration pay lower cash dividends". Isn’t this normal during a crisis period, for firms to pay less dividends?.

This is not always true because some studies such as Sawicki (2009) and Mili et al (2017) prove that the best corporate governance companies pay higher dividends in crisis periods and vice versa.

-   As indicated in our work, we used the fixed effect regression model according to the Hausman test. Moreover, this model helps reduce the effect of unobserved heterogeneity due to time-invariant omitted variables. 

-   We divided the study period into three sub-periods, namely: period before crisis, period during crisis and period after crisis (Model 3).Within the framework of our study we considered the Tunisian revolution that occurred in 2011 as a crisis period.

-   To make the results easy to follow, we reorganized the tables as requested.

6/ Conclusions: The authors refer mainly to the use of their research for Tunisia, but in order for the paper to be interesting for a wider audience they need to discuss the results and their relevance for other countries and companies.

Response:

Many thanks for your comment. The new conclusion generalizes the implications for a global audience as you proposed. It reads now as follows. 

Therefore, the results of this study seem of great interest to illustrate the important disciplinary role that the nature of ownership structure and board of directors’ characteristics plays in a context of concentrated ownership. In this respect, this research paper is useful for Tunisian listed companies’ board of directors to gain a better understanding of the influential factors of dividend policy. Besides, our results are very useful, particularly for managers and shareholders in order to rationalize and to adjust their dividend payout strategies during crises period. Indeed, having an efficient dividend scheme during periods of crises allows the company to be more competitive and maximizes shareholders’ profits. Subsequently, since ownership structure and board of directors’ characteristics are considered to be factors that disrupt the dividend decision, our findings can also be utilized by managers and financial analysts in the field of economics and finance to make knowledge able and effective decisions in order to improve the firm’s performance. Furthermore, these findings appear to be relevant given that in recent years investors have shown growing interest in ownership structure and board of directors’ characteristics (Graham et al. 2020; Suwaidan and Khalaf 2020).

In addition to the practical contributions that have just been mentioned, this study also brings theoretical contributions to the field of financial research by exploring this field in Tunisia and more generally in an emerging economy. The findings of this study contribute to existing theories and may be used for future researchers to conduct further analysis and to draw comparisons of the results from other studies on the topic, in different geographical, economic and political settings. Thus, the present paper is addressed not only to a Tunisian or a global audience within the research field of finance, but also to firm managers and stakeholders. Furthermore, it contributes with novel insights about the impact of effective corporate governance mechanisms on the firm’s dividend payout decisions.

Reviewer 2 Report

This study examines how ownership structure (ownership concentration and CEO duality) affects the dividend distribution policy with a sample of Tunisia. This study attempts to answer the question as to how the DDP can be derived in a firm and I found the approach the authors took interesting. However ,I found a few things to be further considered for publication. Let me discuss them briefly as shown below:

  1. Theory and hypotheses

I acknowledge that the relationship between ownership concentration as well as CEO duality and DDP is inconclusive. Given that, the authors may want to address this puzzle first with a sound theoretical framework or with contextualized rationales. For the former one, while the authors have already presented the underlying mechanism regarding managerial discretion, they may need to elaborate the theoretical framework to cover all the hypotheses comprehensively. It would be best if the theoretical framework can reconcile the competing hypotheses. Or, the authors may want to present a succinct theory which can support the empirical results. This echoes with the latter one (i.e. contextualized rationales). The Tunisian contexts could be further explained to justify the hypotheses.

  1. Empirical models

I understand that the crisis is an important moderator to affect the relationship between ownership structure and DDP. Thereupon, the variable of Crisis should be in the all model as a control variable. In Model 1 of Table 4, the coefficient of the ownership concentration is significant and negative while the model 2 shows its effect is significant and positive – inconsistent results. I don’t think this turned-over sign is a moderation effect of Crisis. The models could have been revisited. Perhaps, Model 1 could not be reported. To clarify the effect of Crisis, the authors may want to remove the observations of the crisis period and see how the results are changed. For model 3, I think the authors may want to conduct diff-in-diff analyses, which are more relevant for this study.  

  1. discussion

The findings should be interpreted based on the Tunisian contexts. Why are the results found in the sample of Tunisia? How can the empirical results be contextualized? Given that this paper argues that the Tunisian contexts reflects an emerging market, do the findings show a consistent result with the other emerging market? If not, why? In the discussion section, the authors may want to address these questions.

I hope these comments are helpful to develop this paper.

Author Response

Dear Editor,

I would like to thank you and the reviewers for your helpful comments and suggestions that were instrumental in dramatically improving our work. Below please find the detailed responses for each comment. All new revisions in the revised version of the manuscript are highlighted in yellow. Also, please consider the latest version of our manuscript as we have made numerous changes following the reviewers’ suggestions. We hope that these revisions meet you’re your expectations and those of the referees.

Should you have any question or additional suggestions, please do not hesitate to contact me.
Best regards 

Dr. Anis El Ammari

 Manuscript ID: ijfs-1374877

Reviewer 2: Comments and Suggestions for Authors

This study examines how ownership structure (ownership concentration and CEO duality) affects the dividend distribution policy with a sample of Tunisia. This study attempts to answer the question as to how the DDP can be derived in a firm and I found the approach the authors took interesting. However, I found a few things to be further considered for publication. Let me discuss them briefly as shown below:

1/Theory and hypotheses

I acknowledge that the relationship between ownership concentration as well as CEO duality and DDP is inconclusive. Given that, the authors may want to address this puzzle first with a sound theoretical framework or with contextualized rationales. For the former one, while the authors have already presented the underlying mechanism regarding managerial discretion, they may need to elaborate the theoretical framework to cover all the hypotheses comprehensively. It would be best if the theoretical framework can reconcile the competing hypotheses. Or, the authors may want to present a succinct theory which can support the empirical results. This echoes with the latter one (i.e. contextualized rationales). The Tunisian contexts could be further explained to justify the hypotheses.

Response:

Thank you for your insightful and constructive comment. Concerning the literature relating to the hypotheses development, we have developed contextualized rationales. The new version contains all new modifications that respond to your suggestion.

2/Empirical models

I understand that the crisis is an important moderator to affect the relationship between ownership structure and DDP. Thereupon, the variable of Crisis should be in the all model as a control variable. In Model 1 of Table 4, the coefficient of the ownership concentration is significant and negative while the model 2 shows its effect is significant and positive – inconsistent results. I don’t think this turned-over sign is a moderation effect of Crisis. The models could have been revisited. Perhaps, Model 1 could not be reported. To clarify the effect of Crisis, the authors may want to remove the observations of the crisis period and see how the results are changed. For model 3, I think the authors may want to conduct diff-in-diff analyses, which are more relevant for this study.

Response: Thank you for your interesting comment.

  • Contrary to what you pointed out in your following remark “In Model 1 of Table 4, the coefficient of the ownership concentration is significant and negative while the model 2 shows its effect is significant and positive”, the coefficient of the ownership concentration variable (in model 1 of Table 4) rather has a positive and significant sign, while the model 2 shows that its effect is significant and negative. - (on the contrary that you indicated in your remark). This change of the sign justifies the moderating effect of Crisis.

  • We have already removed the observation of the crisis period in Model 1 (Model 1 without crisis). Then in a second step, we introduced this variable in model 2 (Model 2 with crisis). As you suggested, this allows us to clarify the effect of Crisis.

 3/ discussion

The findings should be interpreted based on the Tunisian contexts. Why are the results found in the sample of Tunisia? How can the empirical results be contextualized? Given that this paper argues that the Tunisian contexts reflect an emerging market, do the findings show a consistent result with the other emerging market? If not, why? In the discussion section, the authors may want to address these questions.

Response:

We thank the reviewer for this suggestion and we agree with this point of view. To this end, within the framework of the findings we tried to interpret the results according to the Tunisian contexts.

Reviewer 3 Report

Thank you for providing me with the opportunity to review your paper. I have enjoyed reading it. The article is well structured, the hypotheses are well founded in the literature. The discussion and conclusion do not need improvement.

I only have minor repairs to make:

  1. It looks like something is missing from figure 1.
  2. The authors should review how citations should be made, as they are not in accordance with the journal's guideline.

I hope my feedback on this paper will help the authors to improve their work.

Author Response

Dear Editor,

I would like to thank you and the reviewers for your helpful comments and suggestions that were instrumental in dramatically improving our work. Below please find the detailed responses for each comment. All new revisions in the revised version of the manuscript are highlighted in yellow. Also, please consider the latest version of our manuscript as we have made numerous changes following the reviewers’ suggestions. We hope that these revisions meet you’re your expectations and those of the referees.

Should you have any question or additional suggestions, please do not hesitate to contact me.

Best regards 

Dr. Anis El Ammari

 Manuscript ID: ijfs-1374877

Reviewer 3: Comments and Suggestions for Authors

Thank you for providing me with the opportunity to review your paper. I have enjoyed reading it. The article is well structured; the hypotheses are well founded in the literature. The discussion and conclusion do not need improvement.

I only have minor repairs to make:

  1. It looks like something is missing from figure 1.
  2. The authors should review how citations should be made, as they are not in accordance with the journal's guideline.

I hope my feedback on this paper will help the authors to improve their work.

Response:

Thank you very much for your valuable comments.

  • We have adjusted the figure to be align with our hypotheses.
  • We agree with the viewpoint of the reviewer regarding the citations. For this purpose, we adjusted the citations in accordance with the journal’s guideline.

Round 2

Reviewer 1 Report

I would like to congratulate the authors for their work on the paper and the changes made, that have improved the manuscript. 

Author Response

Thank you for your comments, I have revised accordingly.

Reviewer 2 Report

The points I raised at the prior review round have been well-addressed. I appreciate for all the efforts the authors had for revising the paper.

Author Response

(The authors gave the same response as above.)
